# Diagnostic performances of pepsinogens and gastrin-17 for atrophic gastritis and gastric cancer in Mongolian subjects

**Ganchimeg Dondov[1,2], Dashmaa Amarbayasgalan[3], Batbold Batsaikhan[1], Tegshjargal Badamjav[1], Batchimeg Batbaatar[4], Baljinnyam Tuvdenjamts[4], Nasanjargal Tumurbat[1], Bayar Davaa[1], Erkhembulgan Purevdorj[1], Bayarmaa Nyamaa[2], Tulgaa Lonjid●[1] ***

**1** Department of Internal Medicine, Institute of Medical Sciences, Mongolian National University of Medical Sciences, Ulaanbaatar, Mongolia, **2** Department of Gastroenterology, School of Medicine, Mongolian National University of Medical Sciences, Ulaanbaatar, Mongolia, **3** Department of Radiology, National Cancer Center, Ulaanbaatar, Mongolia, **4** Central Research Laboratory, Institute of Medical Sciences, Mongolian National University of Medical Sciences, Ulaanbaatar, Mongolia

* tulgaa.ims@mnums.edu.mn

**Data Availability Statement:** All CSV data and README files are available from the Dryad

## Abstract

In Mongolia, gastric cancer morbidity and mortality are high, and more than 80 percent of cases are diagnosed at an advanced stage. This study aimed to evaluate pepsinogens (PGIs) and gastrin-17 (G-17) levels and to determine the diagnostic performances for gastric cancer and chronic atrophic gastritis among Mongolian individuals. We enrolled a total of 120 subjects, including gastric cancer (40), atrophic gastritis (40), and healthy control (40), matched by age (±2) and sex. Pepsinogen I (PGI), Pepsinogen II (PGII), G-17, and *H. pylori* IgG levels were measured using GastroPanel ELISA kit (Biohit, Helsinki, Finland). Also, PGI to PGII ratio (PGR) was calculated. For atrophic gastritis, when the optimal cut-off value of PGI was ≤75.07 ng/ml, the sensitivity and specificity were 75% and 50%, respectively; when the optimal cut-off value of PGR was ≤6.25, sensitivity and specificity were 85% and 44.7%, respectively. For gastric cancer, when the optimal cut-off value of PGI was ≤35.25 ng/ml, the sensitivity and specificity were 47.2% and 86.8%, respectively; when the optimal cut-off value of PGR was ≤5.27, sensitivity and specificity were 75% and 60.5%, respectively. Combinations of biomarkers with risk factors could improve diagnostic accuracy (AUC for atrophic gastritis 74.8, 95% CI 64.0–85.7, p<0.001; AUC for gastric cancer 75.5, 95% CI 64.2–86.8, p<0.001). PGI, PGR biomarkers combined with the risk of age, family history of gastric cancer, and previous gastric disease could not be an alternative test for upper endoscopy but might be a supportive method which is identifying individuals at medium- and high risk of gastric cancer and precancerous lesions who may need upper endoscopy.

database (https://doi.org/10.5061/dryad.
6wwpzgn1t).

**Funding:** Tulgaa Lonjid and Erkhembulgan
Purevdorj received grant and supervised study.The
current study was funded by Science and
Technology Foundation, Ulaanbaatar, Mongolia
(Grant No: SHUSs2018/15). URL: http://stf.mn/en/.
The funders had no role in study design, data
collection and analysis, decision to publish, or
preparation of the manuscript.

**Competing interests:** The authors have declared
that no competing interests exist.

## Introduction

Gastric cancer is still a major health problem worldwide, despite a dramatic reduction in incidence and mortality rates [1]. In Mongolia, gastric cancer is ranking second after liver cancer and it is increasing in the last decade [2]. According to data of the International Agency for Research on Cancer, Mongolia had the highest rate (100'000:32.5) of gastric cancer and leads to mortality (100'000:24.6) of gastric cancer [3]. Many epidemiological studies have revealed a strong association between *H. pylori* infection and gastric cancer. Previous studies have shown that the prevalence of *H. pylori* infection is high among the population of Mongolia [4]. Besides, over 80% of gastric cancer cases are diagnosed in the late stage in our country [5]. But a screening program has not introduced to decrease the gastric cancer rate. According to the study, early detection of gastric cancer could reduce death related gastric cancer by 30–65% [6]. In our country, we performed a gastroduodenoscopy for screening and histological evaluation to diagnose gastric cancer. These methods are an effective diagnostic modality for gastric diseases; however, invasive and cause discomfort, making it an undesirable procedure for patients. Thus, there is a demand to introduce a non-invasive, easy-to-use early detection method for screening to general the population. Gastric cancer is the end of a long and multi-step process, including atrophic gastritis, intestinal metaplasia, low- and high-grade dysplasia [7]. Accordingly, we considered that atrophy is the key condition of gastric cancer and monitoring atrophic gastritis is a preventive measure against gastric cancer. In some developed countries, *H. pylori* IgG, pepsinogens (PGs), and gastrin-17 (G-17) have been studied as non-invasive serological evaluation of gastric cancer and precancerous gastric lesions and have been suggested a variety of cut-off values [8–11]. Human PGs, which are protein-digestive enzymes secreted as proenzymes by chief cells, classified as pepsinogen I (PGI) and pepsinogen II (PGII). PGs may function as a marker of the functional and morphologic status of the gastric mucosa [12,13]. Gastrins are synthesized by endocrine G cells of the stomach and stimulates the secretion of gastric acid by the parietal cells. Gastric acid is necessary for the conversion of inactive pepsinogen to active pepsin. G-17 and G-34 are classical gastrins, G-17 is more prevalent in antral mucosa and G-34 predominates in duodenum. Also, gastrins allow proliferation, inhibit apoptosis and support migration of gastric epithelial cells [14]. Based on these physiologies, the loss of glands in atrophy would decrease PGs and G-17 levels. Therefore, we aimed to evaluate serum PGIs and G-17 levels and to determine the diagnostic performances for gastric cancer and chronic atrophic gastritis among Mongolian individuals.

## Materials and methods

### Study subjects

This study enrolled a total of 120 subjects who attended the gastrointestinal endoscopy at the National Center of Cancer of Mongolia between January 2019 and October 2019. There were 40 gastric cancer patients enrolled before surgery and other therapies. Besides, there were 40 chronic atrophic gastritis patients and 40 healthy controls enrolled. Gastric cancer and chronic atrophic gastritis were confirmed by gastroscopy and pathological examination; healthy controls were without obvious disease by basic tests, and healthy mucosa by gastroscopy. Subjects of three groups were matched by age (±2) and sex. Exclusion criteria were followed: age <18, pregnancy, recent use of proton pump inhibitor or $H_2$ receptor blockers, history of *H. pylori* eradication within three months, history of gastric surgery or malignancy of cancers. After the exclusion of 6 subjects, 36 with gastric cancer, 40 with chronic atrophic gastritis, and 38 healthy subjects were included. Table 1 summarizes the baseline characteristics of the study population. This study was performed by the Helsinki Declaration and all subjects signed

**Table 1. Baseline characteristics of the study population.**

| Variables | Total (n = 114) | Healthy control, (n = 38) | Atrophic gastritis (n = 40) | Gastric cancer (n = 36) | *p value* |
|---|---|---|---|---|---|
| **Age, mean (SD)**[*] | 59.98 (10.88) | 59.87 (11.62) | 58.73 (10.79) | 61.50 (10.26) | - |
| **Gender, male (%)**[*] | 60 (52.6) | 20 (52.6) | 21 (52.5) | 19 (52.8) | - |
| **Family history of gastric cancer (%)** | 24 (21.1) | 7 (18.4) | 4 (10.0) | 13 (36.1) | *0.018* |
| **Previous history of gastrointestinal diseases (%)** | 35 (30.7) | 6 (15.8) | 10 (25.0) | 19 (52.8) | *0.002* |
| ***H. pylori* IgG >30EIU (%)** | 67 (58.8) | 24 (63.2) | 22 (55.0) | 21 (58.3) | *0.764* |
| **PGI ng/ml, median (min, max)** | 59.38 (4.23, 223.94) | 74.32 (14.65, 223.94) | 56.52 (4.23, 209.28) | 46.94 (6.52, 212.67) | *0.067* |
| **PGII ng/ml, median (min, max)** | 13.11 (3.73, 41.77) | 13.32 (6.24, 39.48) | 11.39 (4.20, 40.75) | 16.60 (3.73, 41.77) | *0.084* |
| **PGR, median (min, max)** | 4.80 (0.58, 13.37) | 5.77 (1.71, 12.87) | 5.03 (0.60,13.73) | 3.76 (0.58, 8.71) | *0.004* |
| **G-17 pmol/l, median (min, max)** | 12.36 (0.11, 78.69) | 13.99 (0.80, 78.69) | 10.64 (0.96–45.80) | 13.41 (0.11, 77.49) | *0.091* |

[*]Age and gender were matched between study groups.

PGI, pepsinogen I; PGII, pepsinogen II; PGR, pepsinogen I to pepsinogen II ratio; G-17, gastrin-17.

informed consent to participate in this study, which was approved by the Ethics Review Committee of the Ministry of Health of Mongolia on August 24, 2018 (Approval №67).

## Gastrointestinal endoscopy

Gastrointestinal endoscopy was performed at the National Cancer Center of Mongolia with accordance national standard MNS5747-1:2007 using endoscope EVIS Exera III. After the 10-hour fast, simethicone solution was used to improve the visibility of the mucosa, followed by 10% lydocaine spray. Endoscopies were initially performed using white light. Subsequently, narrow-band imaging was activated, if any further evaluation was required. At least four biopsies were obtained from the corpus, antrum, and lesions detected macroscopically and by narrow-band imaging. The biopsies were transferred into 4% formalin buffer and sent to the Department of Pathology for pathological examination. The diagnosis was confirmed based on the gastroscopy and pathological examinations by a single expert, respectively.

## Measurement of serum biomarkers using GastroPanel

PGI, PGII, G-17, and *H. pylori* IgG levels were measured using GastroPanel enzyme-linked immunosorbent assay kit (Biohit, Helsinki, Finland). To obtain more accurate analysis results, the biomarkers concentration was used for the average value of the results of a triplicate analysis repeated twice. The fasting blood samples were collected into an EDTA tube from all subjects. The blood samples were centrifuged at 2000 rpm for 10 minutes, and the supernatant was stored at -70°C freezer until testing. Before the assay, the samples were diluted with diluent buffer for the assays following the manufacturer's package insert: 1:5 for G-17, 1:20 for PGI and PGII, and 1:400 for H. pyloriIgG. The plasma concentrations of PGI, PGII, G-17, and *H. pylori* IgG were determined by following protocol. First, the blank solutions, calibrators, controls and diluted samples were pipetted into microplate wells at a volume of 100 μl. Each individuals sample were pipetted into 3 microplate wells. The microplates were incubated at room temperature for 60 minutes with shaking at 750 rpm. Microplate strips were automatically washed three times with 350 μl of diluted buffer and gently tapped on a clean towel. Subsequently, 100 μl of specific conjugate solutions were pipetted into each microplate wells and incubated at room temperature for 60 minutes with shaking at 750 rpm. Microplate strips were automatically washed using a BIOBASE-EL10A reader three times with 350 μl of diluted buffer and gently tapped on a clean towel. After that, 100 μl of substrate solutions were pipetted into each microplate wells and incubated for 30 minutes at ambient temperature avoiding

exposure to light. Finally, 100 μl of stop solutions were pipetted into microplate wells. The absorbance of the microplate wells was measured at 450 nm using a BIOBASE-EL10A microplate reader (Biobase Biodustry, Shandong, China). Also, PGI to PGII ratio (PGR) was calculated.

## Statistical analysis

All statistical analyses were performed by SPSS version 26.0 (SPSS, Chicago, IL, USA). Categorical variables were presented as numbers and proportions and differences were assessed using the Chi-square test. Plasma levels of biomarkers were presented as medians and differences assessed using Kruskal-Wallis Test. The diagnostic accuracy and cut-off values were assessed by ROC curves and the Youden index, with evaluations of sensitivity, specificity, positive predictive value, negative predictive value, positive likelihood ratio, and negative likelihood ratio. Additionally, we evaluated all subjects by giving one point to each of the age $\leq 40$, positive family history of gastric cancer, positive previous disease history, PGI $\leq 75.07$ ng/ml, PGR $\leq 6.25$, or two point to each of PGI $\leq 35.25$ ng/ml, and PGR $\leq 5.27$, with score ranging from 0 to 7 to predict gastric cancer and atrophic gastritis risks. Differences with $p < 0.05$ considered to be statistically significant.

## Results

### Subjects and biomarkers levels

Baseline characteristics and levels of PGI, PGII, G-17 and PGR of study subjects are presented in Table 1. Median age of the subjects was 62 (min 27, max 80), 52.6% (n = 60) were male. Proportions of family history of gastric cancer and previous history of gastric disease were significantly higher in the gastric cancer group compared with atrophic gastritis and healthy control groups. Biopsy of gastric cancer patients yielded the following findings: adenocarcinoma 75.0% (n = 27), mucinous carcinoma 11.1% (n = 4), poorly cohesive 8.3% (n = 3), tubular carcinoma 2.8% (n = 1), papillary carcinoma 2.8% (n = 1). The median of PGI was 74.32 ng/ml (14.65 to 223.94) for healthy controls, 56.52 ng/ml (4.23 to 209.28) for atrophic gastritis and 46.94 ng/ml (6.52 to 212.67) for gastric cancer patients. The PGI level was significantly decreased in gastric cancer and atrophic gastritis groups as compared to the healthy control ($p < 0.05$, $p < 0.05$) (Fig 1A). The median of PGII was 13.32 ng/ml (6.24 to 39.48) for healthy controls, 11.39 ng/ml (4.20 to 40.75) for atrophic gastritis and 16.60 ng/ml (3.73 to 41.77) for gastric cancer patients. And the median of G-17 was 13.99 pmol/l (0.80 to 78.89) for healthy controls, 11.39 pmol/l (0.96 to 45.80) for atrophic gastritis and 13.41 pmol/l (0.11 to 77.49) for gastric cancer patients. There were no significant differences in the PGII and G-17 levels between study groups (Fig 1B and 1D). The median of PGR was 5.77 (1.71 to 12.87), 5.03 (0.60 to 13.73), and 3.76 (0.58 to 8.71) for healthy controls, atrophic gastritis, and gastric cancer patients, respectively. The PGR was significantly lower in the gastric cancer group compared with the healthy control ($p < 0.01$) (Fig 1C).

### Diagnostic performance of the biomarkers for atrophic gastritis

The corresponding ROC curves of PGI and PGR were developed to predict atrophic gastritis, and AUC were 65.2 (95% CI 53.0–77.3, $p < 0.05$) and 62.7 (95% CI 50.1–75.3, $p < 0.05$), respectively (Fig 2A). The results calculated diagnostic values of PGI, PGR, G-17 and combinations of biomarkers for atrophic gastritis have summarized in Table 2. When the optimal cut-off value of PGI was $\leq 75.07$ ng/ml, the sensitivity and specificity were 75% and 50%, respectively. Also, when the optimal cut-off value of PGR was $\leq 6.25$, sensitivity and specificity were 85% and 44.7%, respectively. Because the AUC was relatively lower, we developed ROC curves for

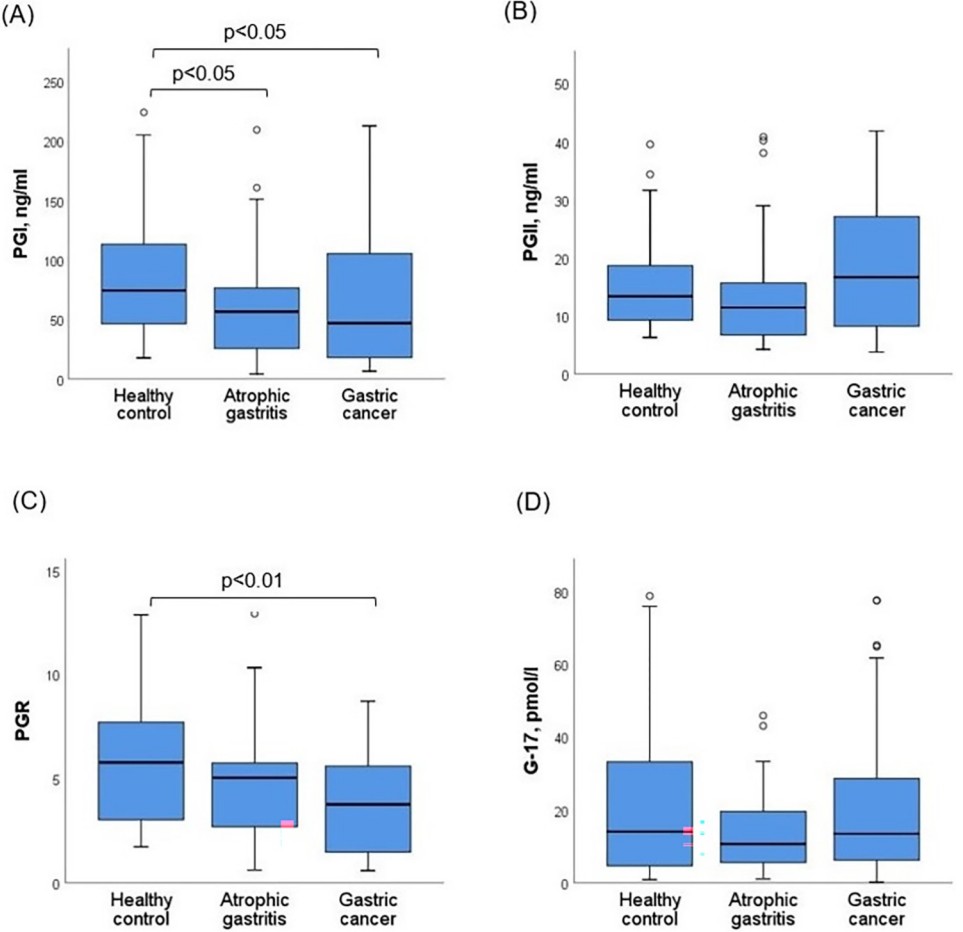

**Fig 1.** Comparisons of (A) PGI, (B) PGII, (C) PGR, (D) G-17 biomarkers levels between healthy control and patients with atrophic gastritis and gastric cancer. The graph indicates the medians and the boxes of 25%-75% quartiles. PGI, pepsinogen I; PGII, pepsinogen II; PGR, pepsinogen I to pepsinogen II ratio; G-17, gastrin-17. *H. pylori* was positive in 67 (58.8%) subjects according to *H. pylori* IgG assay and there was no difference between study groups. We estimated PGI, PGII, PGR and G-17 levels between *H. pylori* IgG negative and positive groups. The median of PGI was 46.11 (4.22 to 188.07) and 70.26 (7.53 to 223.94), the median of PGII was 8.40 (3.73 to 41.77) and 15.78 (4.43 to 40.75) for *H. pylori* IgG negative and positive subjects, respectively. The PGI and PGII levels were higher in *H. pylori* positive subjects than in *H. pylori* negative subjects (p<0.01), while there was no difference in PGR and G-17 levels between them. The median of PGR was 4.62 (0.60 to 13.37) and 4.85 (0.58 to 12.86), the median of G-17 was 10.52 (0.11 to 77.49) and 13.69 (1.02 to 78.69) for *H. pylori* IgG negative and positive subjects, respectively.

combinations of biomarkers. But AUC of PGI and/or PGR combination was not significant to predict atrophic gastritis. However, level of G-17 was not different among study groups, AUC of G-17 (cut-off value ≤23.42) combined with PGI and/or PGR was 70.3 (95% CI 58.4–82.1, p<0.01) with 80% sensitivity and 60.5% specificity (Table 2).

## Diagnostic performance of the biomarkers for gastric cancer

The corresponding ROC curves of PGI and PGR were developed to predict gastric cancer, and AUC were 64.3 (95% CI 51.3–77.2, p<0.05) and 71.6 (95% CI 69.6–82.8, p<0.01), respectively (Fig 2B). The results calculated diagnostic values of PGI, PGR and combinations of biomarkers for gastric cancer have presented in Table 2. When the optimal cut-off value of PGI was ≤35.25 ng/ml, the sensitivity and specificity were 47.2% and 86.8%, respectively. When the

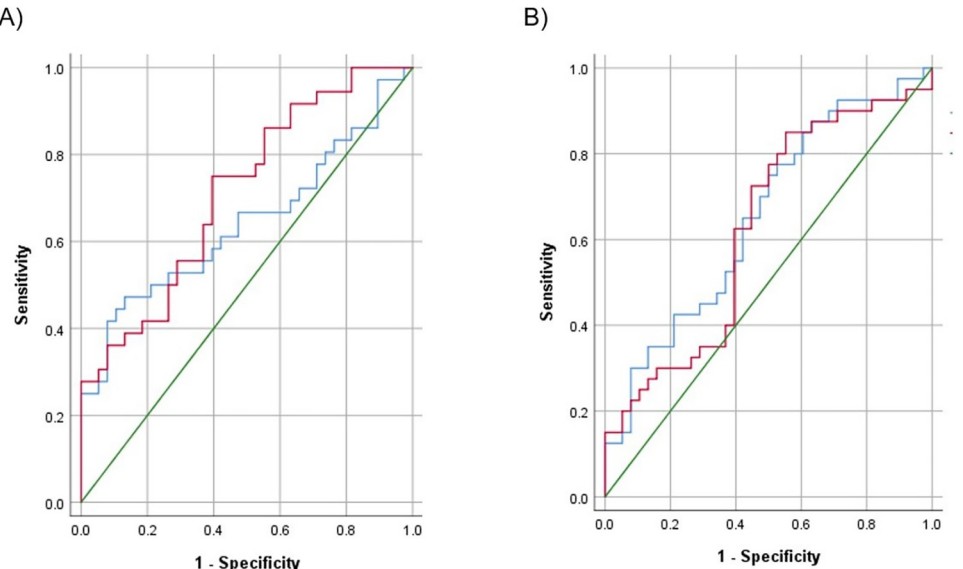

**Fig 2.** Receiver Operating Characteristic (ROC) curves for diagnosing gastric cancer (A) and atrophic gastritis (B). The blue line indicates PGI and the red line indicates PGR. A green line is reference line. PGI, pepsinogen I; PGR, pepsinogen I to pepsinogen II ratio.

optimal cut-off value of PGR was ≤5.27, sensitivity and specificity were 75% and 60.5%, respectively. Also, we developed ROC curves for combinations of biomarkers, to increase AUC. The AUC of PGI and/or PGR combination was 69.2 (95% CI 56.9–81.4, p<0.01), with sensitivity of 77.7% and specificity of 60.5% (Table 2).

## Scoring system to predict risk of atrophic gastritis and gastric cancer

In addition, we evaluated all subjects by giving one point to each of the age ≤40, positive family history of gastric cancer, positive previous gastric disease history, PGI ≤75.07 ng/ml, PGR

**Table 2. Diagnostic performance of biomarkers for detection of gastric cancer and atrophic gastritis.**

| Biomarker | Outcome* | Cut-off | Sens % | Spec % | PPV % | NPV % | LR+ | LR- | AUC (95% CI) | p value |
|---|---|---|---|---|---|---|---|---|---|---|
| **PGI** | AG | ≤75.07 | 75.0 | 50.0 | 61.2 | 65.5 | 1.5 | 0.7 | 65.2 (53.0–77.3) | *0.021* |
| | GC | ≤35.25 | 47.2 | 86.8 | 77.3 | 63.5 | 3.6 | 0.3 | 64.3 (51.3–77.2) | *0.035* |
| **PGR** | AG | ≤6.25 | 85.0 | 44.7 | 61.8 | 73.9 | 1.5 | 0.7 | 62.7 (50.1–75.3) | *0.048* |
| | GC | ≤5.27 | 75.0 | 60.5 | 58.7 | 67.9 | 1.9 | 0.5 | 71.6 (69.6–82.8) | *0.002* |
| **G-17** | AG | ≤23.42 | 85.0 | 39.5 | 59.6 | 71.4 | 1.4 | 0.7 | 57.3 (44.3–70.3) | *0.267* |
| **PGI and/or PGR** | AG | PGI≤75.07 PGR≤6.25 | 95.0 | 26.3 | 57.5 | 83.3 | 1.3 | 0.8 | 60.7 (48.0–73.3) | *0.105* |
| | GC | PGI≤35.25 PGR≤5.27 | 77.7 | 60.5 | 65.1 | 74.2 | 2.0 | 0.5 | 69.2 (56.9–81.4) | *0.005* |
| **G-17 and PGI and/or PGR** | AG | G-17≤23.42 PGI≤75.07 PGR≤6.25 | 80.0 | 60.5 | 68.1 | 74.2 | 2.0 | 0.5 | 70.3 (58.4–52.1) | *0.002* |

*Outcome of AG was analyzed between healthy controls (n = 38) and atrophic gastritis (n = 40) subjects; outcome of GC was analyzed between healthy controls (n = 38) and gastric cancer (n = 36) subjects.

AG, atrophic gastritis; GC, gastric cancer; PGI, pepsinogen I; PGR, pepsinogen I to pepsinogen II ratio; G-17, gastrin-17; Sens, sensitivity; spec, specificity; PPV, positive predictive value; NPV, negative predictive value; LR+, positive likelihood ratio, LR-, negative likelihood ratio, AUC, area under the curve.

**Table 3. Prevalence rate and risks of atrophic gastritis and gastric cancer by risk category.**

| Score category | Healthy control n (%) | Atrophic gastritis | | Gastric cancer | |
|---|---|---|---|---|---|
| | | n (%) | OR (95% CI) | n (%) | OR (95% CI) |
| Low-risk | 19 (50.0%) | 6 (15.0%) | ref | 4 (11.1%) | ref |
| Medium-risk | 12 (31.6%) | 17 (42.5%) | 4.49 (1.38–14.58) | 11 (30.6%) | 4.35 (1.13–16.85) |
| High-risk | 7 (18.4%) | 17 (42.5%) | 7.69 (2.16–27.43) | 21 (58.3%) | 14.25 (3.60–56.43) |

≤6.25, or two point to each of PGI ≤35.25 ng/ml, and PGR ≤5.27, with score ranging from 0 to 7. As score increased, the risk of atrophic gastritis or gastric cancer increased. Scores 0 to 2, 3 to 4, 5 to 7 were classified into three categories, corresponding to low-, medium-, high-risk, respectively. According to classification, 29 (25.4%) subjects were classified into the low-risk category, 40 (35.1%) subjects into medium-risk category, and 45 (39.5%) subjects into high-risk category. For the atrophic gastritis patients, 17 (42.5%) were classified into medium-risk category (OR 4.49, 95% CI 1.38–14.58) and 17 (42.5%) were classified into high-risk category (OR 7.69, 95% CI 2.16–27.43). Whereas, 11 (30.6%) patients with gastric cancer were classified into medium-risk category (OR 4.35, 95% CI 1.13–16.85), 21 (58.3%) were classified into high-risk category (OR 14.25, 95% CI 3.60–56.43) (Table 3).

## Discussion

In Mongolia, gastric cancer morbidity and mortality are high, and more than 80 percent of cases are diagnosed at an advanced stage. So, it's recommended to have a gastrointestinal endoscopy annually for individuals who are over 40 years. But the endoscopic and histological examinations are invasive and unpleasant for individuals. There is a demand to introduce a non-invasive, easy-to-use early detection methods for screening to the general population. The purpose of this case-control study was to evaluate PGI, PGII, G-17 level and PGR and determine diagnostic performances for atrophic gastritis and gastric cancer compared to healthy controls. Because some studies showed that male gender was associated with higher PGI than female [15,16] and positive correlation between age and PGI or PGII [17], we matched our study subjects by age (±2) and sex.

According to the results, atrophic gastritis and gastric cancer patients were associated with a low level of PGI and PGR. Previous studies have shown that the low level of PGI, PGR with *H. pylori* examination can predict gastric cancer and precancerous lesions with a variety of cut-off values [4,8,18,19]. In our study, the optimal cut-off value of PGI was ≤75.07 ng/ml with 75% sensitivity, 50% specificity and ≤35.25 ng/ml with 47.2% sensitivity, 86.8% specificity for atrophic gastritis and gastric cancer, respectively. These findings were approximate to other studies which have been suggested a PGI cut off ≈70 ng/ml and ≈30 ng/ml for atrophic gastritis and gastric cancer, respectively [8,13,18]. PGI sensitivity for gastric cancer (47.2%) in our result is consistent with previous studies noted that sensitivity is lower (36.8%-62.3%) than the assessment of gastric atrophy [20–22]. In contrast, PGR has better sensitivity for the assessment of atrophic gastritis and gastric cancer in this study (85% and 75.0%) and previous studies (73.5–87.1%) [20,22]. PGR cut-off values (≤6.25 for atrophic gastritis, ≤5.27 for gastric cancer), were quite higher in our result than some studies suggested [8,18]. But a similar outcome had reported in a study by Cao Q et al (2007), for the best discrimination of atrophic gastritis, the cut-off values of PGI and PGR were 82.3 microg/L and 6.05, respectively [19].

Previous study revealed that high level of serum G-17 (>15 pmol/L) was significantly associated with increased risk of atrophic gastritis in healthy population. But the progression of stomach diseases, the diagnostic strength of serum G-17 for atrophic gastritis declined in more

advanced situations, such as gastric cancer [23]. Our finding showed that the AUCs of the models with G-17 for atrophic gastritis was higher than without G-17; however, G-17 level had not difference between study groups. Therefore, we recommend including G-17 in the screening of precancerous lesions and prediction models for accurate risk stratification of gastric cancer.

The age of patients, positive family history of gastric cancer, and positive previous gastric disease history are known as risk factors for gastric cancer from our study. So, we combined these risk factors with biomarkers to increase diagnostic accuracy. Our finding highlighted the combinations of biomarkers with risk factors could improve diagnostic accuracy (AUC for atrophic gastritis 74.8, 95% CI 64.0–85.7, p<0.001; AUC for gastric cancer 75.5, 95% CI 64.2–86.8, p<0.001). But, the sensitivity and specificity of biomarkers in our finding indicated that these non-invasive biomarkers cannot be perfect alternative methods to endoscopic examination for the diagnosis of gastric cancer and precancerous lesions. However, these are might be valuable screening markers for the high-risk population, who may need upper endoscopy. Cai Q et al (2019) comprised seven variables, including age, sex, PGR, G-17 level, *H. pylori* infection, pickled food and fried food, with scores from 0 to 25 to stratify high-risk population in China. According their results, the observed prevalence rates of gastric cancer in the derivation cohort at low-risk (≤11), medium-risk (12–16) or high-risk (17–25) group were 1.2%, 4.4% and 12.3%, respectively (p<0.001) [24]. In this study, we created a risk prediction scoring system with a score ranging from 0 to 7, based on variables age, family history of gastric cancer, prior disease history, PGI and PGR levels. Our findings revealed that medium-risk (3–4 score) or high-risk (5–7 score) categories have more prevalence of patients with atrophic gastritis and gastric cancer. So, we recommended that patients who are classified into medium-risk or high-risk category have to investigate further examination, such as upper endoscopy.

In this study, *H. pylori* IgG was not associated with atrophic gastritis and gastric cancer. This finding can be explained by the high prevalence of *H. pylori* infection among the population of our country [25]. Moreover, our data showed that the PGI and PGII levels were higher in *H. pylori* positive subjects, while there was no difference in PGR. The PGI and PGII levels were markedly increased in *H. pylori* infection, in contrast PGI level was declined in atrophic mucosal change and cancer [26]. Based on this findings, PGR might be more valuable biomarker to distinguish gastritis.

Different population-based screening strategies currently being adopted successfully in Korea, Japan, and high incidence regions of China and Taiwan for gastric cancer [27]. Mongolia leads to the morbidity and mortality of gastric cancer and 80% of gastric cancer cases are diagnosed in the late stage [5]. Unfortunately, a screening program has not introduced to decrease the gastric cancer rate. Therefore, there is an urgent need to introduce effective screening methods to decreasing gastric cancer incidence and mortality. Recently, endoscopy is a predicting method for gastric cancer and precancerous lesions in our country. The sensitivity and specificity of endoscopic screening are varying in different countries. For example, in countries such as South Korea and Japan, where the incidence of gastric cancer is high, the sensitivity of the endoscopy is more than 80% [28,29]. The use of endoscopic screening method for gastric cancer has several limitations in our country, such as insufficient patient enrollment due to invasiveness, poor supply of endoscopic devices for all regions, and deficiency well-trained endoscopist to meet the increased demand. Therefore, these biomarkers, combined with the risk of age, family history, and previous gastric disease might be considered supportive method for the mass screening of gastric cancer and precancerous lesions in our country.

Our study has several limitations. First, due to the small number of subjects, subjects with atrophic gastritis and gastric cancer have not classified into different clinical classifications.

Previous studies reported that a low level of serum PGI and PGR more related to corpus atrophy and diffuse type gastric cancer [4,30]. Therefore, a study with the large number of subjects is needed. Second, *H. pylori* infection evaluated only using antibody assay. According to some studies, serum PGII and PGR level is associated with *H. pylori* infection status and its eradication [27,31].

In conclusion, PGI, and PGR biomarkers could not be an alternative test for upper endoscopy but might be a supportive method. The scoring system, based on PGI, PGR, risk of age, family history of gastric cancer and previous history of gastric disease could identify individuals with medium- and high-risk gastric cancer and precancerous lesions who may need upper endoscopy.

## Acknowledgments

We would like to thank National Cancer Center of Mongolia, especially for Dr Amarbat Baatarnum, Dr Serjbayar Ganbold, Dr Chinzorig Munkhjargal, and Dr Erkhembayar Enkhbat for enrolling their patients in this study.

## Author Contributions

**Formal analysis:** Ganchimeg Dondov, Dashmaa Amarbayasgalan.

**Funding acquisition:** Erkhembulgan Purevdorj, Tulgaa Lonjid.

**Investigation:** Ganchimeg Dondov, Dashmaa Amarbayasgalan, Tegshjargal Badamjav, Batchimeg Batbaatar, Baljinnyam Tuvdenjamts, Nasanjargal Tumurbat, Bayar Davaa.

**Methodology:** Ganchimeg Dondov, Dashmaa Amarbayasgalan, Batbold Batsaikhan, Tegshjargal Badamjav, Nasanjargal Tumurbat, Bayarmaa Nyamaa, Tulgaa Lonjid.

**Project administration:** Batbold Batsaikhan, Tulgaa Lonjid.

**Resources:** Bayar Davaa.

**Software:** Ganchimeg Dondov, Batbold Batsaikhan.

**Supervision:** Erkhembulgan Purevdorj, Bayarmaa Nyamaa, Tulgaa Lonjid.

**Validation:** Batchimeg Batbaatar, Baljinnyam Tuvdenjamts.

**Writing – original draft:** Ganchimeg Dondov, Tegshjargal Badamjav, Nasanjargal Tumurbat.

**Writing – review & editing:** Batbold Batsaikhan, Erkhembulgan Purevdorj, Bayarmaa Nyamaa, Tulgaa Lonjid.

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
