## [Decision Letter · Decision Letter 0]

5 Jan 2022

PONE-D-21-19578Diagnostic performances of serum Pepsinogens and Gastrin-17 for atrophic gastritis and gastric cancer in Mongolian subjectsPLOS ONE

Dear Dr. Lonjid,

Thank you for submitting your manuscript to PLOS ONE. After careful consideration, we feel that it has merit but does not fully meet PLOS ONE’s publication criteria as it currently stands. Therefore, we invite you to submit a revised version of the manuscript that addresses the points raised during the review process.

We look forward to receiving your revised manuscript.

Kind regards,

Muhammad Tarek Abdel Ghafar, M.D

Academic Editor

PLOS ONE

Journal Requirements:

Additional Editor Comments:

The sample size appears to be small. Please add a justification for the sample size used in this study.

Reviewers' comments:

Reviewer's Responses to Questions

**Comments to the Author**

1. Is the manuscript technically sound, and do the data support the conclusions?

Reviewer #1: Yes

Reviewer #2: Yes

Reviewer #3: Partly

Reviewer #4: No

2. Has the statistical analysis been performed appropriately and rigorously? 

Reviewer #1: Yes

Reviewer #2: Yes

Reviewer #3: No

Reviewer #4: No

3. Have the authors made all data underlying the findings in their manuscript fully available?

Reviewer #1: Yes

Reviewer #2: Yes

Reviewer #3: No

Reviewer #4: No

4. Is the manuscript presented in an intelligible fashion and written in standard English?

Reviewer #1: Yes

Reviewer #2: Yes

Reviewer #3: Yes

Reviewer #4: No

5. Review Comments to the Author

Reviewer #1: This manuscript entitles "Diagnostic performances of serum pepsinogens and Gastrin-17 for atrophic gastritis and gastric cancer in Mongolian subjects" by Dondov G et. al have assessed the efficacy of serum pepsinogens and gastrin-17 levels as biomarkers for gastric cancer in Mongolia. It is a well-reported phenomenon that serum pepsinogens and Gastrin-17 exhibit the potential to be used as diagnostic markers for gastric cancer. Given the discomfort associated with endoscopy, serum markers can pave the way for an easy diagnosis of the disease. This article provide novel insights about the levels of pepsinogens and gastric-17 in Mongolian population which has not been studied yet. The only serious drawback or limitation of the study is its small population size, which the authors have mentioned.

I would recommend this paper for publication.

Reviewer #2: The topic is important and the manuscript is well-structured.

-Please provide a reference for the score used to combine the biomarkers with other risk factors.

-The sensitivity and specificity is relatively low, even after applying the score. I think this is not sufficient to reach this conclusion.

Reviewer #3: This is an interesting paper in some regards. The authors selection of a matched case study was good, the study arms are balanced and well controlled.

Unfortunately, there are a number of issues with the study results and interpretation, listed below:

1. The data plotted in Figure 1 demonstrate a very substantial overlap between the control population and the two disease populations, Atrophic Gastritis and Gastric Cancer. The graph of the PGR biomarker (pepsinogen ratios) shows a believable trend towards decrease with advancing disease. This is known from the literature. The data presented here and in Table 2, although showing a "significant" statistical difference in the populations, demonstrate that the serum assay cannot distinguish the benign from malignant disease. The data in both the figure and table demonstrate a very low clinical sensitivity for separation of the two disease populations from the healthy population.

2. The authors proposed a combined biomarker using 4 scoring factors. This is an interesting approach. However, not enough data was provided on the application of the 4 factor biomarker score on this study. A Table and Figure covering the population data set for all patients for the proposed marker should be included.

3. There is a Figure 2 in the manuscript, showing ROC plots for the two biomarkers selected for analysis, PGI (serum pepsinogen I) and PGR (serum pepsinogen I-II ratios). The plots appear to show a somewhat better performance of the markers when comparing gastric cancer to normal vs gastric atrophy to normal. The tabular analysis demonstrates that the serum marker levels will not distinguish the two diseases. Therefor the authors need to provide a cogent argument on how the serum assay would actually be useful in the clinic and what would be the benefit of a low sensitivity biomarker assay that cannot distinguish benign from malignant disease; the authors point out the lack of available endoscopy facilities in Mongolia but also point out that only pathology workup can distinguish benign from cancerous lesions; does this mean that endoscopy is unlikely to be useful or will endoscopy replace the serum assay in most circumstances due to better capability of separating normal from diseased (benign plus cancer) populations.

Other items to be addressed:

Line 80; Solution is misspelled

Line 96; change wording to "following the manufacturer's package insert"

Line 105; automatically washed, this probably means with a plate washer; please specify including model used

Lines 138-144; demonstrate that there is not an adequate rationale for including the PG-II and G17 measurements in the data analyses

Line 162-164; given that there is no measurable difference across the populations for PG-II and G17, there inclusion to make the data more statistically significant is not justified

Table 2; there are two columns labelled as positive likelihood ratio. presumable one should be labelled negative; this reviewer does not understand what the authors mean by a positive likelihood ratio. is this just that the biomarker is positive or is this a probability of disease. if the latter, is it restricted to cancer or is it either benign or cancer.

Line 225; the high background incidence of H pylori in Mongolia is a critical factor in understanding this study; this sentence should be moved to the introduction.

Lines 252-272; this paragraph is unnecessarily long and redundant, and should be reworked.

Line 280 is unsupported by the data presented, and yet is likely the most important outcome of this study if true.

Reviewer #4: Review: PONE-D-21-19578

The authors reported the levels of PG1, PGR, Gastrin-17 as diagnostic markers for gastric cancer in the

Mongolian subjects. The authors used the GastroPanel (ELISA, BioHit) to measure the levels of these

three molecules and H. pylori IgG. GastroPanel has been extensively used to identify healthy gastric

mucosa, H. pylori infection, anacidic stomach, atrophic gastritis and guide risk patients to gastroscopy.

These four molecules have been extensively studied in the Asian population due to high prevalence of

gastric cancer in this geography. The authors reported the H. pylori positive (58.8%) in the study groups

but did not incorporate H. pylori results as a risk factor for chronic atrophic gastritis or gastric cancer. H.

pylori IgG, age, gender, and smoking status have been reported as risk factors for gastric cancer. The

authors did not cite either old or recent publications describing these four molecules for gastric cancer,

for examples:

1) Cao et al. (2007) in Journal of Digestive Disease, “Screening of atrophic gastritis and gastric cancer by

serum pepsinogen, gastrin-17 and Helicobacter pylori immunoglobulin G antibodies”, where H. pylori

IgG results were also incorporated in the analyses with PGI and PGR. These authors reported the

significant increase in G-17 in the gastric cancer group.

2) Reviewed publications, Kim N. & Jung C. (2010) Gut Liver “The role of serum pepsinogen in the

detection of gastric cancer”.

3) Wang & Chen (2020) Scientific Reports, “Prevalence of atrophic gastritis in southwest China and

predictive strength of serum gastrin-17: A cross-sectional study (SIGES)”

4) Shen et al. (2021) Gastroenterology Research and Practice “The diagnostic value of serum gastin-17

and pepsinogen for gastric cancer screening in eastern China”.

A few highlights for authors to consider:

Line 80, “simethicone silution” should be “solution”.

Line 83, the authors described the collection of biopsies, but nowhere in the manuscript described the

results from these samples.

Line 92 of the manuscript “The fasting blood samples were collected in an EDTA tube from all subjects.”

The presence of EDTA as anticoagulant in the tube is for plasma collection, not serum, as the authors

used “serum” throughout the manuscript.

Line 224, “In this study, H. pylori IgG was not associated with atrophic gastritis and gastric cancer”.

Nowhere in the manuscript the authors described the analyses for PGI, PGR, and G-17 relative to H.

pylori IgG positive and negative subgroups.

6. PLOS authors have the option to publish the peer review history of their article (what does this mean?). If published, this will include your full peer review and any attached files.

Reviewer #1: No

Reviewer #2: **Yes: **Dina M. El-Guindy

Reviewer #3: No

Reviewer #4: No

---

## [Author Response · Author response to Decision Letter 0]

29 Mar 2022

We would like to appreciate for editor. We are grateful your consideration of our manuscript and for giving us the opportunity to improve it. Also, we thank to the reviewers for their careful reading of our manuscript. Reviewers comments and suggestions were very helpful and valuable in improving the article. Updated cover letter and two versions of the manuscript are attached, one where all the changes have been highlighted blue texts and another one without any highlights. 

We hope that our revision to the manuscript will facilitate the decision to publish our article in your journal. 

Thank you again consideration.

---

## [Decision Letter · Decision Letter 1]

30 May 2022

PONE-D-21-19578R1Diagnostic performances of Pepsinogens and Gastrin-17 for atrophic gastritis and gastric cancer in Mongolian subjectsPLOS ONE

Dear Dr. Lonjid,

Thank you for submitting your manuscript to PLOS ONE. After careful consideration, we feel that it has merit but does not fully meet PLOS ONE’s publication criteria as it currently stands. Therefore, we invite you to submit a revised version of the manuscript that addresses the points raised during the review process.

We look forward to receiving your revised manuscript.

Kind regards,

Muhammad Tarek Abdel Ghafar, M.D

Academic Editor

PLOS ONE

Reviewers' comments:

Reviewer's Responses to Questions

**Comments to the Author**

1. If the authors have adequately addressed your comments raised in a previous round of review and you feel that this manuscript is now acceptable for publication, you may indicate that here to bypass the “Comments to the Author” section, enter your conflict of interest statement in the “Confidential to Editor” section, and submit your "Accept" recommendation.

Reviewer #2: All comments have been addressed

Reviewer #3: All comments have been addressed

2. Is the manuscript technically sound, and do the data support the conclusions?

Reviewer #2: Yes

Reviewer #3: Yes

3. Has the statistical analysis been performed appropriately and rigorously? 

Reviewer #2: Yes

Reviewer #3: Yes

4. Have the authors made all data underlying the findings in their manuscript fully available?

Reviewer #2: Yes

Reviewer #3: Yes

5. Is the manuscript presented in an intelligible fashion and written in standard English?

Reviewer #2: Yes

Reviewer #3: Yes

6. Review Comments to the Author

Reviewer #2: Please provide a reference for similar scoring.

Why the score variables are different regarding AG and GC?

The score used need to be clearly mentioned in the methods section.

The aim and title need to be modified as the authors didn't evaluated pepsinogen and gastrin 17 alone.

Reviewer #3: (No Response)

7. PLOS authors have the option to publish the peer review history of their article (what does this mean?). If published, this will include your full peer review and any attached files.

Reviewer #2: No

Reviewer #3: No

---

## [Author Response · Author response to Decision Letter 1]

9 Jul 2022

Reviewer 2’s comments:

Comment 1: Please provide a reference for similar scoring. Why the score variables are different regarding AG and GC? The score used need to be clearly mentioned in the methods section.

Response: We strongly agree with your comment that score variables are different. So, we conducted new analysis and created new scoring system combined with atrophic gastritis and gastric cancer. We have added our results under additional subtitle “Scoring system to predict risk of atrophic gastritis and gastric cancer” (Line 206-222). Cai Q et al (2019) comprised seven variables, including age, sex, PGR, G-17 level, H. pylori infection, pickled food and fried food, with scores from 0 to 25 to stratify high-risk population in China (ref 24). Thank you for your valuable 

Comment 2: The score used need to be clearly mentioned in the methods section.

Response: According to your comment, we have mentioned about the scoring system in the method section (Line 120-124). Thank you for pointing out this.

---

## [Editor Report · Decision Letter 2]

12 Jul 2022

PONE-D-21-19578R2Diagnostic performances of Pepsinogens and Gastrin-17 for atrophic gastritis and gastric cancer in Mongolian subjectsPLOS ONE

Dear Dr. Lonjid,

Thank you for submitting your manuscript to PLOS ONE. After careful consideration, we feel that it has merit but does not fully meet PLOS ONE’s publication criteria as it currently stands. Therefore, we invite you to submit a revised version of the manuscript that addresses the points raised during the review process.

We look forward to receiving your revised manuscript.

Kind regards,

Muhammad Tarek Abdel Ghafar, M.D

Academic Editor

PLOS ONE

Additional Editor Comments (if provided):

Please resubmit the revised manuscript and point-by-point response files while deleting any redundant files from the previous revisions.
---

## [Author Response · Author response to Decision Letter 2]

17 Jul 2022

We have deleted previous manuscript and response files and resubmitted the new revised manuscript and response to reviewer's files. Also, we have uploaded and checked our figure files to the Preflight Analysis and Conversion Engine (PACE) digital diagnostic tool to PLOS requirements. Thank you for your suggestions.

---

## [Decision Letter · Decision Letter 3]

1 Aug 2022

PONE-D-21-19578R3Diagnostic performances of Pepsinogens and Gastrin-17 for atrophic gastritis and gastric cancer in Mongolian subjectsPLOS ONE

Dear Dr. Lonjid,

Thank you for submitting your manuscript to PLOS ONE. After careful consideration, we feel that it has merit but does not fully meet PLOS ONE’s publication criteria as it currently stands. Therefore, we invite you to submit a revised version of the manuscript that addresses the points raised during the review process.

We look forward to receiving your revised manuscript.

Kind regards,

Muhammad Tarek Abdel Ghafar, M.D

Academic Editor

PLOS ONE

Journal Requirements:

Reviewers' comments:

Reviewer's Responses to Questions

**Comments to the Author**

1. If the authors have adequately addressed your comments raised in a previous round of review and you feel that this manuscript is now acceptable for publication, you may indicate that here to bypass the “Comments to the Author” section, enter your conflict of interest statement in the “Confidential to Editor” section, and submit your "Accept" recommendation.

Reviewer #2: (No Response)

2. Is the manuscript technically sound, and do the data support the conclusions?

Reviewer #2: Partly

3. Has the statistical analysis been performed appropriately and rigorously? 

Reviewer #2: Yes

4. Have the authors made all data underlying the findings in their manuscript fully available?

Reviewer #2: Yes

5. Is the manuscript presented in an intelligible fashion and written in standard English?

Reviewer #2: Yes

6. Review Comments to the Author

Reviewer #2: The authors didn't provide a reference for the used score.

Have the authors proposed this score?

If so, then the conclusion needs to be modified as they are not evaluating a well established score.

7. PLOS authors have the option to publish the peer review history of their article (what does this mean?). If published, this will include your full peer review and any attached files.

Reviewer #2: **Yes: **Dina M. El-Guindy

---

## [Author Response · Author response to Decision Letter 3]

30 Aug 2022

Dear Editor

Thank you for giving us the opportunity to submit a revised draft of the manuscript “Diagnostic performances of serum Pepsinogens and Gastrin-17 for atrophic gastritis and gastric cancer in Mongolian subjects” for publication in the PLOS ONE. We appreciate the time and effort that you and reviewers dedicated to providing feedback on our manuscript and are grateful for the generous comments on and valuable improvements to our paper. Following reviewer’s suggestion, we have made a minor revisions to the manuscript. Those changes are highlighted within the manuscript. Changes to the manuscript are shown in blue. 

We hope the revised version is now suitable for publication and look forward to hearing from you. 

Here, we have answered comment below. 

Reviewer 2’s comments:

Comment: The authors didn't provide a reference for the used score. Have the authors proposed this score? If so, then the conclusion needs to be modified as they are not evaluating a well established score.

Response: We strongly agree with your comment. We have added reference score of medium- and high risk in round bracket to make it more clear (Line 274-276). Also, we have modified our conclusion according to your comment that conclusion needs to be proposed scoring system (Line 309-313). We considered that our scoring system could identify individuals who may need upper endoscopy. The use of endoscopic screening method for gastric cancer has several limitations in our country, such as insufficient patient enrollment due to invasiveness, poor supply of endoscopic devices for all regions, and deficiency well-trained endoscopists to meet the increased demand. Therefore, these biomarkers, combined with the risk of age, family history, and previous gastric disease might be considered supportive method for the mass screening of gastric cancer and precancerous lesions in our country. Thank you for your useful comments. We believe that our manuscript are improving with your valuable suggestions. 

We look forward to hearing from you and responding to any further questions and comments you may have.

---

## [Decision Letter · Decision Letter 4]

8 Sep 2022

Diagnostic performances of Pepsinogens and Gastrin-17 for atrophic gastritis and gastric cancer in Mongolian subjects

PONE-D-21-19578R4

Dear Dr. Lonjid,

We’re pleased to inform you that your manuscript has been judged scientifically suitable for publication and will be formally accepted for publication once it meets all outstanding technical requirements.

Kind regards,

Muhammad Tarek Abdel Ghafar, M.D

Academic Editor

PLOS ONE

Additional Editor Comments (optional):

Reviewers' comments:

Reviewer's Responses to Questions

**Comments to the Author**

1. If the authors have adequately addressed your comments raised in a previous round of review and you feel that this manuscript is now acceptable for publication, you may indicate that here to bypass the “Comments to the Author” section, enter your conflict of interest statement in the “Confidential to Editor” section, and submit your "Accept" recommendation.

Reviewer #2: All comments have been addressed

2. Is the manuscript technically sound, and do the data support the conclusions?

Reviewer #2: Yes

3. Has the statistical analysis been performed appropriately and rigorously? 

Reviewer #2: Yes

4. Have the authors made all data underlying the findings in their manuscript fully available?

Reviewer #2: Yes

5. Is the manuscript presented in an intelligible fashion and written in standard English?

Reviewer #2: Yes

6. Review Comments to the Author

Reviewer #2: (No Response)

7. PLOS authors have the option to publish the peer review history of their article (what does this mean?). If published, this will include your full peer review and any attached files.

Reviewer #2: No

---

## [Editor Report · Acceptance letter]

6 Oct 2022

PONE-D-21-19578R4 

Diagnostic performances of Pepsinogens and Gastrin-17 for atrophic gastritis and gastric cancer in Mongolian subjects 

Dear Dr. Lonjid:

I'm pleased to inform you that your manuscript has been deemed suitable for publication in PLOS ONE. Congratulations! Your manuscript is now with our production department. 

Kind regards, 

on behalf of

Prof Muhammad Tarek Abdel Ghafar 

Academic Editor

PLOS ONE